# Multistability of the Vibrating System of a Micro Resonator

Yijun Zhu 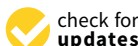 and Huilin Shang *

School of Mechanical Engineering, Shanghai Institute of Technology, Shanghai 201418, China; zyjmain@163.com
* Correspondence: suliner60@hotmail.com

**Abstract:** Multiple attractors and their fractal basins of attraction can lead to the loss of global stability and integrity of Micro Electro Mechanical Systems (MEMS). In this paper, multistability of a class of electrostatic bilateral capacitive micro-resonator is researched in detail. First, the dynamical model is established and made dimensionless. Second, via the perturbating method and the numerical description of basins of attraction, the multiple periodic motions under primary resonance are discussed. It is found that the variation of AC voltage can induce safe jump of the micro resonator. In addition, with the increase of the amplitude of AC voltage, hidden attractors and chaos appear. The results may have some potential value in the design of MEMS devices.

**Keywords:** micro resonator; fractal; multistability; safe jump; hidden attractor; chaos; basin of attraction



## 1. Introduction

Multistability, i.e., the coexistence of multiple attractors, is a common dynamical phenomenon in MEMS/NEMS [1,2]. Based on it, there are many applications such as MEMS-based memory [3] and switches [4]. In addition, considering the loss of global stability that multistability may trigger, there are some devices that should avoid the appearance of multiple attractors in their vibrating systems, such as filters [5], microvalves [6], and micro-relays [7]. As one of the fastest developing MEMS products [8], electrostatic micro-resonators should assure that the resonators undergo periodic vibration whose amplitudes vary continuously with the driven voltages. However, in practical applications of electrostatic micro-resonators [9], there are many complex dynamic behaviors such as multistability [10,11], quasi-periodic motion [12] and periodic-n motion [13], chaos [14,15], and pull-in instability.

It is of great significance to study the multistability and necessary conditions for inducing it either for avoiding this phenomenon or making use of it. Thus, multistability of vibrating systems of micro resonators has been studied experimentally and numerically during these decades [16]. Via experiments, Mohammadreza investigated the dynamic response of an electrostatic micro-actuator in the vicinity of the primary resonance and the parametric one [17]. Siewe et al. [18] studied the vibration of a double-side MEMS resonator numerically and found the variation of the driven voltage could induce the coexistence of chaos and quasi-periodic motions. Shang et al. [19] found the coexisting chaos and dynamical pull-in in the vibrating system of a single-side electrostatic micro sensor. Haghighi et al. [20] found the coexisting periodic-n motion and the chaotic motion of micromechanical resonators with electrostatic forces on both sides, and then discussed the global bifurcation of its vibrating system by approximately expressing its homoclinic orbits as the ones of a typical duffing equation. When amplifying signals of a nanomechanical duffing resonator, Almog et al. [21] found that multistability was an interesting dynamical phenomenon of nonlinear systems and could be explored for many applications. Gusso et al. [22] studied chaos of a typical micro/nanoelectromechanical beam resonator with two-sided electrodes experimentally and observed multiple attractors in a significant region of the relevant parameter space, involving periodic and chaotic attractors.

By applying cell-mapping method to depict the basins of attraction for all the attractors, they also found that the basin boundaries were fractal under certain conditions of the excitations, indicating that the attractors are strongly intermingled. Liu et al. [23] applied the method of multiple scales (MMS) to analyze the multiple periodic motions induced by the local bifurcation, and used the Melnikov method to predict necessary conditions for chaos and its control. The corresponding numerical results were also presented by the basins of attraction and spectrum diagrams. Angelo et al. [24] investigated the effect of the linear and nonlinear stiffness terms and damping coefficients on dynamical behaviors of a microelectromechanical resonator and controlled the chaotic motion by forcing it into an orbit obtained analytically via the harmonic balance method. However, most study concentrated on describing or observing the phenomenon itself rather than studying its mechanism, which is still not that clear yet.

To this end, we consider a typical electrostatic driven bilateral capacitive micro-resonator and study the possible multistability and its mechanism in its vibrating system. The paper is organized as follows. In Section 2, the dynamical model is constructed and made dimensionless. In Sections 3 and 4, two different cases for coexisting multiple periodic attractors, fractal basins of attraction, and other complex attractors of the systems are discussed both theoretically and numerically. In Section 5, the conclusions are presented.

## 2. Dynamical Model

We choose to study a class of bilateral micro resonator whose simplified diagram is shown in Figure 1. The driven forces on the resonator are electrostatic ones between the moving electrode and the fixed electrode [25]. The driven voltage in Figure 1 is the combination of alternate current (AC) and direct current (DC) actuation. In the figure, $x$ is the vertical displacement of the moving electrode at moment $t$, $d$ the initial gap width between the moving electrode and each fixed one, $V_b$ the DC bias voltage, $V_{AC}sin\Omega t$ the AC voltage where $V_{AC}$ is the amplitude and $\Omega$ the frequency. Suppose that the amplitude of the AC voltage $V_{AC}$ is much lower than the bias DC voltage $V_b$, i.e., $V_{AC} \ll V_b$. According to the Second Law of Newton, the vibrating system of the moving electrode can be expressed as a nonlinear system as follows:

$$m\frac{d^2x}{dt^2} + c\frac{dx}{dt} + k_1x + k_2x^3 = \frac{C_0}{2(d-x)^2}(V_b + V_{AC}sin\Omega t)^2 - \frac{C_0V_b^2}{2(d+x)^2} \tag{1}$$

where $m$ represents the effective lumped mass of the moving electrode, $k_1$ its linear mechanical stiffness, $k_2$ its cubic nonlinear stiffness, $c$ the damping coefficient, $C_0$ the initial capacitance of the parallel-plate structure.

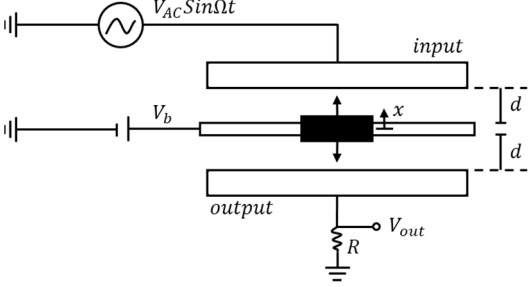

**Figure 1.** Simplified diagram of a bilateral MEMS resonator.

Introducing the following dimensionless variables

$$\omega_0 = \sqrt{\frac{k_1}{m}}, \omega = \frac{\Omega}{\omega_0}, \mu = \frac{c}{m\omega_0}, \alpha = \frac{k_2d^2}{m\omega_0^2}, \beta = \frac{C_0V_b^2}{2k_1d^3}, \gamma = \frac{V_{AC}}{V_b}, T = \omega_0t, u = \frac{x}{d}, \dot{u} = \frac{du}{dT} \tag{2}$$

and substituting Equation (2) into Equation (1), one can obtain that

$$\ddot{u} + \mu\dot{u} + u + \alpha u^3 = \frac{\beta}{(1-u)^2}(1 + \gamma sin\omega T)^2 - \frac{\beta}{(1+u)^2} \tag{3}$$

which is a dimensionless system. Since in the original system (1), the viscous damping coefficient of air $c$ is very tiny, and $V_{AC} \ll V_b$, the parameters $\mu$ and $\gamma$ in (3) will be both small and can be considered as perturbed parameters. Thus, considering $\mu = 0$ and $\gamma = 0$ in Equation (3), one has the unperturbed system that can be expressed as below:

$$\dot{u} = v, \ \dot{v} = -u - \alpha u^3 + \frac{\beta}{(1-u)^2} - \frac{\beta}{(1+u)^2}. \tag{4}$$

Letting the right side of Equation (4) be zero, one can determine equilibria of the dimensionless system (3). Equation (5) is a Hamilton system with the Hamiltonian

$$H(u,v) = \frac{1}{2}v^2 + \frac{1}{2}u^2 + \frac{\alpha}{4}u^4 - \frac{\beta}{1-u} - \frac{\beta}{1+u} + 2\beta \tag{5}$$

and the function of potential energy (P.E.)

$$V(u) = \frac{1}{2}u^2 + \frac{\alpha}{4}u^4 - \frac{\beta}{1-u} - \frac{\beta}{1+u} + 2\beta. \tag{6}$$

Concerning Equation (4), the number of the equilibria, and the shapes and positions of the possible potential wells of the unperturbed system (4) depend on the parameters $\alpha$ and $\beta$. The same as in [20], the values of the parameters in the system (1) are given by:

$$m = 5 \times 12^{-12} \text{ kg}, c = 5 \times 12^{-8} \text{ kg/s}, \ k_1 = 5 \ \mu\text{N}/\mu\text{m}, \ k_2 = 15 \ \mu\text{N}/\mu\text{m}^3, \ d = 2\mu\text{m}, \ C_0 = 1.875 \times 10^{-18} \text{ mF}. \tag{7}$$

Accordingly, in system (4), $\alpha = 12$.

Different equilibria and potential energy diagrams of the unperturbed system under different values of the parameter $\beta$ can be seen in Figure 2. It shows that there are three P.E. poles when $\beta$ = 0.211, five P.E. poles when $\beta$ increases to 0.338, and only one P.E. pole when $\beta$ increases to 0.6. Under different values of $\beta$, the potential wells and unperturbed orbits are shown in Figure 3. When $\beta$ = 0.211, there are three equilibria (two non-trivial equilibria are saddles and the origin is a center) as well as one well surrounded by heteroclinic orbits (see Figure 3a). As $\beta$ increases to 0.338, there will be five equilibria among which two non-trivial equilibria $S_1$ ($-0.196339$,0) and $S_2$ (0.196339,0) are centers of the two wells surrounded by homoclinic orbits; the other three equilibria are unstable. When $\beta$ = 0.6, no wells or non-trivial equilibria of the unperturbed system (4) exist. The P.E. poles in Figure 2 correspond to the fixed points shown in Figure 3. Therefore, according to Equations (2) and (7), when the structural parameters are fixed, the number of centers will depend on the value of DC bias voltage $V_b$: when the DC bias voltage is very low, there will be a center of the system (4) as well as a stable point attractor of the system (3) without AC voltage. Under a higher DC bias voltage, there may be two centers of the system (4). As is well known, periodic vibration can often be attributed to the perturbation of the centers. Since the number and the location of the centers in Figure 3a,b are totally different, the mechanism for the possible multiple periodic attractors of the vibrating system of the micro-resonator can be different as well. Therefore, in Sections 3 and 4, we discuss the different mechanism of multi-stability for these two different cases, i.e., the only center (the origin) and the two non-trivial centers, respectively.

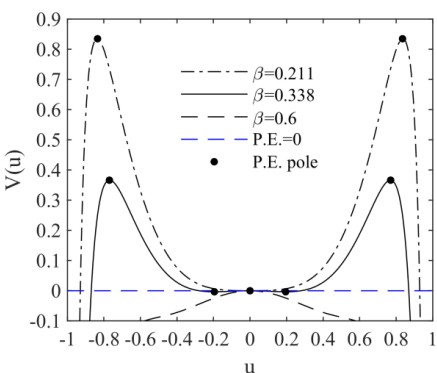

**Figure 2.** Potential energy of the unperturbed system (4) under different values of parameter $\beta$.

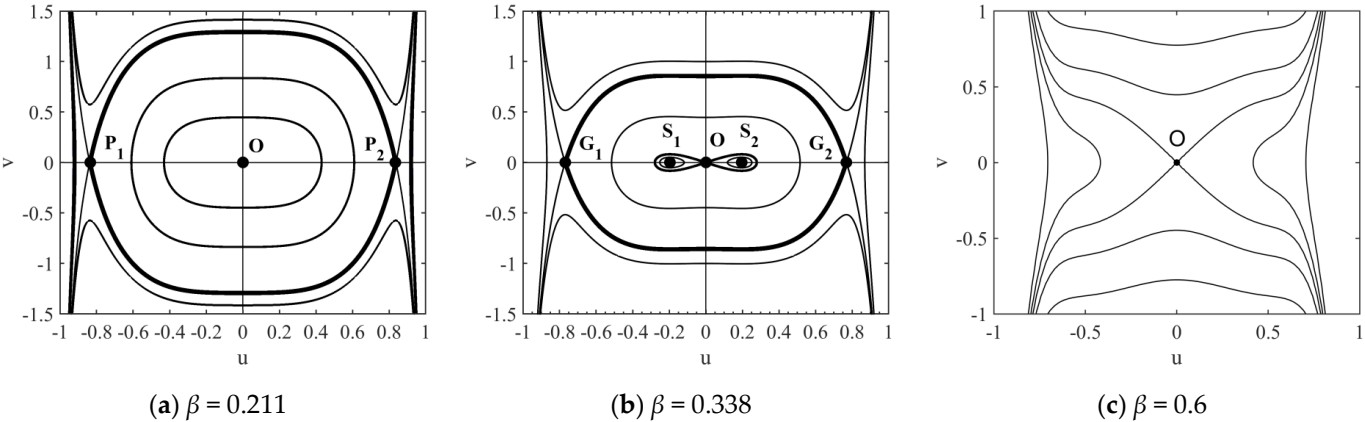

**(a)** $\beta = 0.211$        **(b)** $\beta = 0.338$        **(c)** $\beta = 0.6$

**Figure 3.** Orbits of the unperturbed system (4) under different values of parameter $\beta$.

## 3. Multiple Periodic Attractors in the Neighborhood of the Origin

Considering the case where the DC bias voltage is low, and the periodic vibration of the microstructure is induced by the perturbation of the only center (see Figure 3a, where $V_b = 3$ V), one may use the Method of Multiple Scales (MMS) to analyze the periodic solutions in the neighborhood of the origin. Expanding the fractional terms of the dimensionless system (3) as Taylor series in the neighborhood of $u = 0$, and neglecting the higher-order-than-three terms of $u$, one has:

$$\ddot{u} + \mu\dot{u} + u + \alpha u^3 = 2\beta\gamma\sin\omega T + 4\beta u + 4u\beta\gamma\sin\omega T + 6u^2\beta\gamma\sin\omega T + 8\beta u^3 + 8u^3\beta\gamma\sin\omega T. \tag{8}$$

As mentioned in Section 2, the values of the parameters $\mu$ and $\gamma$ in the above system are small; one can introduce a small parameter $\varepsilon$ satisfying $0 < \varepsilon \ll 1$, and can re-scale the two parameters in the system (8) as:

$$\mu = \varepsilon^2\widetilde{\mu}, \ \gamma = \varepsilon^2\widetilde{\gamma}. \tag{9}$$

Then Equation (8) becomes

$$\ddot{u} + \widetilde{\omega}^2 u = -\varepsilon^2\widetilde{\mu}\dot{u} + 2\varepsilon^2\beta\widetilde{\gamma}\sin\omega T + 4u\varepsilon^2\beta\widetilde{\gamma}\sin\omega T + 6u^2\varepsilon^2\beta\widetilde{\gamma}\sin\omega T - P_1 u^3 + 8u^3\varepsilon^2\beta\widetilde{\gamma}\sin\omega T. \tag{10}$$

where

$$\widetilde{\omega}^2 = 1 - 4\beta, \ P_1 = \alpha - 8\beta. \tag{11}$$

To apply MMS, one may rescale some terms in the system (10) that

$$\omega = \widetilde{\omega} + \varepsilon\sigma, \ u = \varepsilon u_1 + \varepsilon^2 u_2 + \cdots, \ \sigma = O(1). \tag{12}$$

and

$$T_i = \varepsilon^i T, D_i = \frac{\partial}{\partial T_i}, \frac{d}{dT} = \sum_{i=0}^{n} \varepsilon^i D_i \ (i = 0, 1, 2, \cdots) \tag{13}$$

Comparing the coefficients of $\varepsilon^1, \varepsilon^2$, and $\varepsilon^3$ in the system (10), respectively, one obtains that

$$\varepsilon^1: \ D_0{}^2 u_1 + \omega^2 u_1 = 0, \tag{14}$$

$$\varepsilon^2: \ D_0{}^2 u_2 + \omega^2 u_2 = -2D_1 D_0 u_1 + 2\omega\sigma u_1 + 2\beta\widetilde{\gamma}\sin\omega T, \tag{15}$$

and

$$\varepsilon^3: \ D_0{}^2 u_3 + \omega^2 u_3 = -2D_1 D_0 u_2 - \widetilde{\mu} D_0 u_1 - D_1{}^2 u_1 + 2u_2\omega\sigma - \sigma^2 u_1 - 2D_2 D_0 u_1 - P_1 u_1{}^3 + 4\beta\widetilde{\gamma} u_1 \sin\omega T. \tag{16}$$

To solve Equation (14), one can assume that

$$u_1 = A_1(T_1, T_2)e^{i\omega T_0} + \overline{A}_1(T_1, T_2)e^{-i\omega T_0}, \tag{17}$$

where

$$A_1 = \frac{a(T_1, T_2)}{2}e^{i\theta(T_1, T_2)}. \tag{18}$$

Substituting Equations (17) and (18) into Equation (15), and eliminating the secular terms of Equation (15), one will have:

$$D_1 A_1 = -\frac{\beta\widetilde{\gamma}}{2\omega} - i\sigma A_1. \tag{19}$$

Solving Equation (15), one may assume:

$$u_2 = A_2(T_2)e^{i\omega T_0} + \overline{A}_2(T_2)e^{-i\omega T_0}. \tag{20}$$

Substituting Equation (20) into Equation (16), and eliminating secular terms of Equation (16), one will obtain:

$$D_2 A_1 = -\frac{\widetilde{\mu}}{2}A_1 + \frac{\beta\widetilde{\gamma}}{2\omega} - \frac{\sigma\beta\widetilde{\gamma}}{4\omega^2} + \frac{3iP_1 A_1^2 \overline{A}_1}{2\omega}. \tag{21}$$

Since

$$\dot{A}_1 \approx D_0 A_1 + \varepsilon D_1 A_1 + \varepsilon^2 D_2 A_1, \tag{22}$$

Substituting Equations (19) and (21) into Equation (22), and expressing it by the original dimensionless parameters of Equation (3), one has:

$$\varepsilon\dot{a} = -\frac{\mu}{2}(\varepsilon a) - P_2\cos\theta, \tag{23}$$
$$(\varepsilon a)\dot{\theta} = -(\omega - \widetilde{\omega})(\varepsilon a) + \frac{3P_1(\varepsilon a)^3}{8\omega} + P_2\sin\theta.$$

where

$$P_2 = \frac{(3\omega - \widetilde{\omega})\beta\gamma}{2\omega^2}. \tag{24}$$

According to Equation (18), it is obvious that the amplitude of the periodic solution $a$ is the function of the time scale $T_1$ where $T_1$ is a one-order term of $\varepsilon$; thus, one can assume the amplitude of the solution u of Equation (10) $\widetilde{a}$ as:

$$\varepsilon a = \widetilde{a}. \tag{25}$$

Letting $\dot{a} = 0$, and $\dot{\theta} = 0$, one can obtain:

$$-\frac{\mu}{2}\widetilde{a} = P_2\cos\theta, \ (\omega - \widetilde{\omega})\widetilde{a} - \frac{3P_1\widetilde{a}^3}{8\omega} = P_2\sin\theta. \tag{26}$$

Eliminating the triangulation function of Equation (26), one can get:

$$\frac{\mu^2}{4}a^2 + \left(\omega - \widetilde{\omega} - \frac{3P_1 a^2}{8\omega}\right)^2 a^2 = \frac{(3\omega - \widetilde{\omega})^2}{4\omega^4}\beta^2\gamma^2. \tag{27}$$

According to Equations (17), (18) and (25), the periodic solution can be expressed as:

$$u \approx \widetilde{a}\cos(\omega T + \theta). \tag{28}$$

To determine the stability of the periodic solutions, one can get the corresponding characteristic equation of the periodic solution based on Equation (23). It shows that the periodic solution will lose its stability when its amplitude $\widetilde{a}$ satisfies:

$$\left(8(\omega - \widetilde{\omega}) - \frac{9\widetilde{a}^2 P_1}{\omega}\right)\left(8(\omega - \widetilde{\omega}) - \frac{3\widetilde{a}^2 P_1}{\omega}\right) \geq 16\mu^2. \tag{29}$$

Based on Equations (26)–(29), the variation of the amplitude of the periodic solutions of the system (3) and their stability with AC voltage is shown in Figure 4 where the frequency and the amplitude of AC voltage are considered as the control parameters in Figure 4a and Figure 4b, respectively. In Figure 4a, where $V_{AC} = 0.01$ V, when $\omega$ is lower than 0.45, there is only one periodic attractor in the system (3) whose amplitude changes continuously with the increase in $\omega$. Comparatively, when $\omega$ ranges from 0.46 to 0.69, the global dynamical behaviors of the system (3) will change to bistable periodic attractors, which can be attributed to Hopf bifurcation. As $\omega$ continues to increase from 0.7, the periodic attractor with the higher amplitude will disappear, only the periodic attractor with the lower amplitude will exist, and its amplitude will decrease continuously with the increase of $\omega$. Similarly, the change in global dynamical behaviors in Figure 4b also shows that in a certain range of $V_{AC}$, there will be two periodic attractors coexisting, which can be due to Hopf bifurcation of the system (3). The accuracy of the theoretical prediction in Figure 4 are verified by the numerical results.

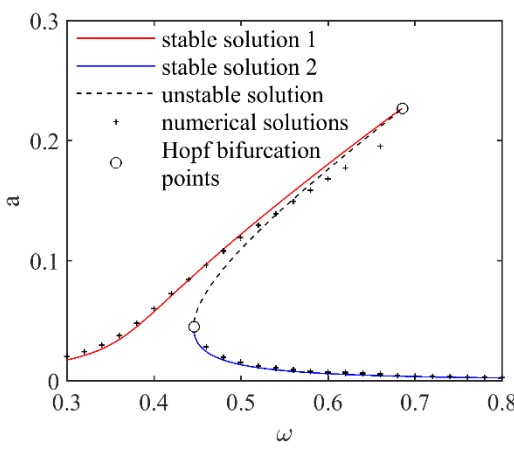

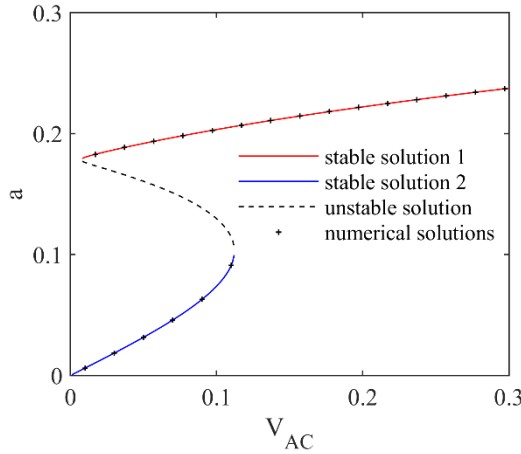

(**a**) Amplitude of the periodic solution vs. $\omega$ when $V_{AC} = 0.01$ V

(**b**) Amplitude of the periodic solution vs. $V_{AC}$ when $\omega = 0.6$

**Figure 4.** Variation of the amplitude of the periodic solution with the change in AC voltage.

Figure 4 demonstrates that the parameters $\omega$ and VAC can induce the coexistence of bistability, meaning that under fixed values of parameters of system (3), different initial conditions may lead to different periodic attractors. Accordingly, it is necessary to classify the basins of attraction for the two different periodic attractors. Here, the 4th order Runge-Kutta approach and the cell-mapping method are applied to depict the basins of attraction of the system (3). The time step is taken as $1/10^2$ of the period of excitation. To investigate the long-term dynamical behaviors, it is supposed that an initial condition will be safe if

the vibration in this initial condition keeps satisfying $|u(T)| < 1$ within $10^5$ excited circles; otherwise, the micro resonator will undergo pull in [20]. The union of all initial conditions leading to the same periodic motion will be the basin of attraction for that attractor which will surely be marked in the same color in the initial plane. The basins of attraction of system (3) are drawn in sufficiently large ranges for the initial position and velocity of the proof mass defined as $|u(0)| < 1$ and $|\dot{u}(0)| < 1.5$ by generating an $200 \times 100$ array of initial points. The change of attractors and the area and nature of their basins of attraction with frequency $\omega$ is shown in Figure 5 where the amplitude of AC voltage $V_{AC}$ is fixed as 0.01 V.

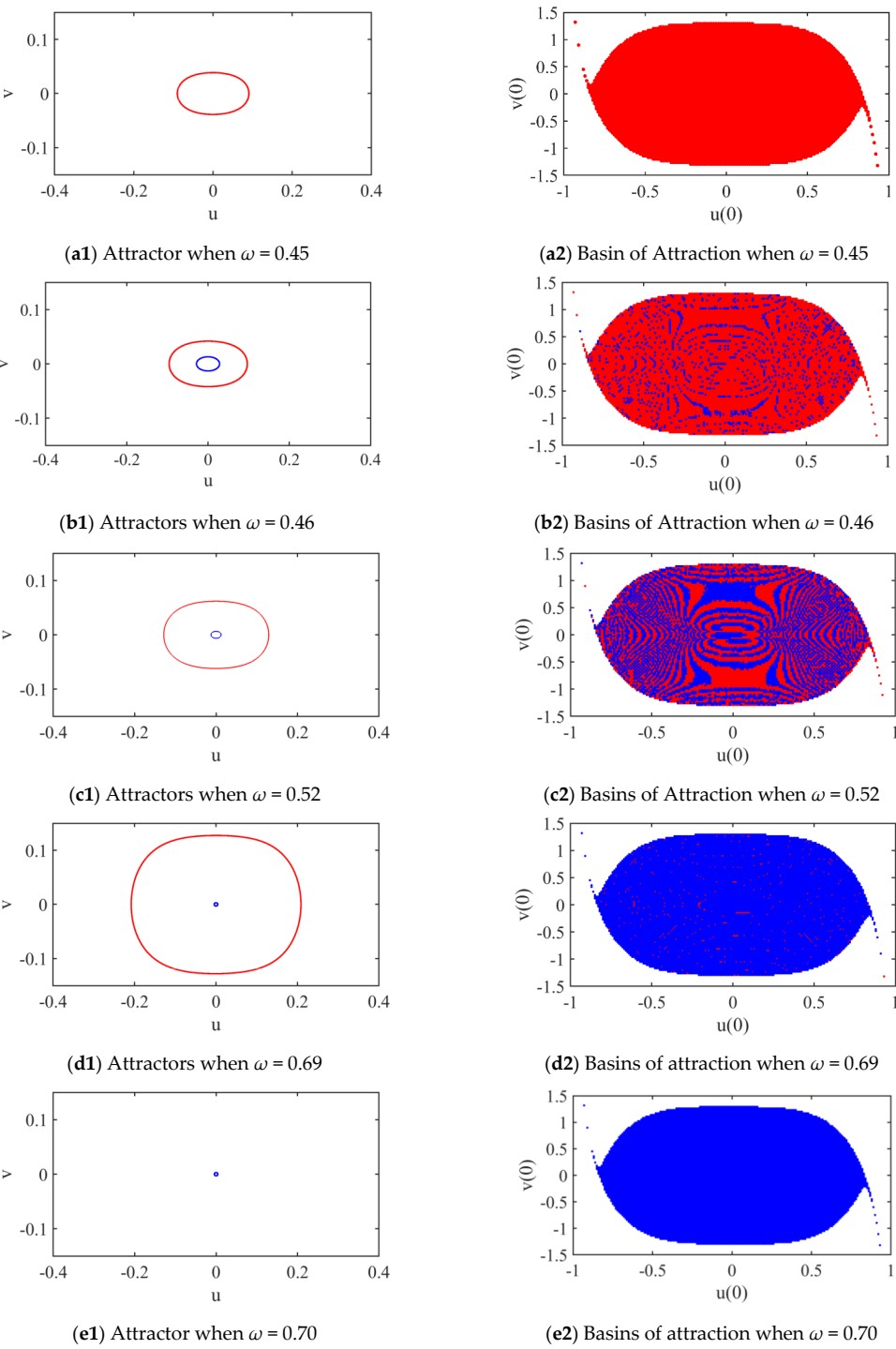

**Figure 5.** Evolution of multiple attractors and their basins of attraction under different values of $\omega$.

According to Figure 5, with the increase in parameter $\omega$, the number of attractors and the boundary of basins of attraction will both change. When $\omega = 0.45$ in system (3), there will only be one periodic attractor whose basin of attraction is comparatively bigger with a smooth boundary (see Figure 5a1,a2). However, with a small increase of $\omega$, i.e., $\omega = 0.45$, the global dynamics are totally different, as shown in Figure 5b1,b2 where two periodic attractors coexist, whose basins of attraction mix each other and are both fractal. It means that the dynamical behavior of system (3) is highly sensitive to initial conditions. In other words, system (3) may undergo a safe jump. A similar phenomenon can be seen in Figure 5c1,c2 under a higher $\omega$. As $\omega$ increases, the basin of attraction of the periodic attractor with the higher amplitude becomes small (see the red regions in Figure 5b2,c2,d2). Specifically, in Figure 5d2, the regions of attraction are almost blue, and there is very little area of basin of attraction for that periodic attractor. As $\omega = 0.70$ (see Figure 5e1,e2), the periodic attractor with the higher amplitude disappears, and there is only the other attractor whose basin of attraction is almost the same as that in Figure 4b, showing that when the frequency $\omega$ increases enough, the periodic attractor with the lower amplitude replaces the initial one.

## 4. Multistability in the Neighborhood of Non-Trivial Equilibria

In this section, the case that the DC voltage is higher, and the periodic vibration of the microstructure is induced by the perturbation of the two non-trivial centers (see Figure 3b) is considered; thus, we set $\beta = 0.338$, i.e., $V_b = 3.8\text{V}$. In addition, we consider the effect of AC voltage on the global dynamics of the system (3). To begin with, setting

$$\varepsilon \hat{u} = u \mp u_c \tag{30}$$

where $u_c$ is the abscissa of the right center (see S$_1$ in Figure 3), rescaling he two parameters $\mu$ and $\gamma$ in the system (3) by Equation (9), expanding the fractional terms of the dimensionless system (3) as a Taylor series in the neighborhood of the non-trivial equilibria and ignoring the higher-order-than-cubic terms of $\hat{u}$, the system (3) becomes

$$\ddot{\hat{u}} = -\varepsilon^2 \widetilde{\mu} \dot{\hat{u}} - \hat{\omega}^2 \hat{u} + \varepsilon Q_1 \hat{u}^2 + \varepsilon^2 Q_2 \hat{u}^3 + \frac{2\varepsilon\beta\widetilde{\gamma}\sin\omega T}{(1 \mp u_c)^2} + \frac{4\varepsilon^2\beta\widetilde{\gamma}\sin\omega T}{(1 \mp u_c)^3}\hat{u} + \frac{6\varepsilon^3\beta\widetilde{\gamma}\sin\omega T}{(1 \mp u_c)^4}\hat{u}^2, \tag{31}$$

where

$$\hat{\omega}^2 = 1 + 3\alpha u_c{}^2 - \frac{4\beta(1 + 3u_c{}^2)}{(1 - u_c{}^2)^3}, \quad Q_1 = \pm 3u_c\left(-\alpha + \frac{8\beta(1 + u_c{}^2)}{(1 - u_c{}^2)^4}\right), \quad Q_2 = -\alpha + \frac{8\beta(1 + 10u_c{}^2 + 5u_c{}^4)}{(1 - u_c{}^2)^5}. \tag{32}$$

To apply the Method of Multiple Scale in Equation (32), one can assume in this equation that:

$$\hat{\omega} = \omega + \varepsilon\hat{\sigma}, \quad \hat{u} = \hat{u}_0 + \varepsilon\hat{u}_1 + \varepsilon^2\hat{u}_2 + \cdots, \quad \hat{\sigma} = O(1). \tag{33}$$

Comparing the coefficients of $\varepsilon^1, \varepsilon^2$ and $\varepsilon^3$, one has:

$$\varepsilon^0 : \ D_0{}^2 \hat{u}_0 + \omega^2 \hat{u}_0 = 0, \tag{34}$$

$$\varepsilon^1 : \ D_0{}^2 \hat{u}_1 + \omega^2 \hat{u}_1 = -2D_1 D_0 \hat{u}_0 + 2\omega\hat{\sigma}\hat{u}_0 + \frac{2\beta\widetilde{\gamma}\sin\omega T}{(1 \mp u_c)^2} + Q_1 \hat{u}_0{}^2, \tag{35}$$

and

$$\varepsilon^2 : \ D_0{}^2 \hat{u}_2 + \omega^2 \hat{u}_2 = -2D_1 D_0 \hat{u}_1 - \widetilde{\mu} D_0 \hat{u}_0 - D_1{}^2 \hat{u}_0 + 2\hat{u}_1\omega\hat{\sigma} - \hat{\sigma}^2 \hat{u}_0 - 2D_2 D_0 \hat{u}_0 + 2Q_1 \hat{u}_0 \hat{u}_1 + Q_2 \hat{u}_0{}^3 + \frac{4\beta\widetilde{\gamma}\sin\omega T}{(1 \mp u_c)^3}\hat{u}_0. \tag{36}$$

One can set the solution of Equation (34) as:

$$\hat{u}_0 = B_1(T_1)e^{i\omega T_0} + \overline{B}_1(T_1)e^{-i\omega T_0}. \tag{37}$$

Substituting Equation (37) into Equation (35), and eliminating the secular terms of Equation (35), one can obtain that:

$$D_1 B_1 = \frac{-\beta\widetilde{\gamma}}{2\omega(1\mp u_c)^2} - i\sigma B_1,$$
$$\hat{u}_1 = -\frac{B_1^2 P}{3\omega^2}e^{i2\omega T_0} - \frac{\overline{B}_1^2 P}{3\omega^2}e^{-i2\omega T_0} + \frac{2B_1\overline{B}_1 P}{\omega^2}. \tag{38}$$

Substituting the equation above into Equation (36) and eliminating its secular terms, one can have:

$$D_2 B_1 = -\frac{\widetilde{\mu}B_1}{2} - \frac{\hat{\sigma}\beta\widetilde{\gamma}}{4\omega^2(1\mp u_c)^2} - i\left(\frac{5Q_1^2}{3\omega^3} + \frac{3Q_2}{2\omega}\right)B_1^2\overline{B}_1. \tag{39}$$

Now setting

$$B_1 = \frac{1}{2}\varepsilon b(T_1, T_2)e^{i\varphi(T_1, T_2)}, \tag{40}$$

considering

$$\dot{B}_1 \approx D_0 B_1 + \varepsilon D_1 B_1 + \varepsilon^2 D_2 B_1, \tag{41}$$

and substituting Equations (38) and (39) into Equation (41), and expressing Equation (41) by the original dimensionless parameters of Equation (3), one has:

$$\dot{b} = \frac{-(3\omega-\hat{\omega})\beta\gamma\cos\varphi}{2\omega^2(1\mp u_c)^2} - \frac{\mu b}{2},$$
$$b\dot{\varphi} = \frac{(3\omega-\hat{\omega})\beta\gamma\sin\varphi}{2\omega^2(1\mp u_c)^2} - (\omega-\hat{\omega})b - \frac{5b^3 Q_1^2}{12\omega^3} - \frac{3b^3 Q_2}{8\omega}. \tag{42}$$

The periodic solution of the system (3) satisfies $\dot{b} = 0$, and $\dot{\varphi} = 0$, i.e.,

$$-\frac{\mu b}{2} = \frac{(3\omega-\hat{\omega})\beta\gamma\cos\varphi}{2\omega^2(1\mp u_c)^2}, \quad (\omega-\hat{\omega})b + \left(\frac{5Q_1^2}{12\omega^3} + \frac{3Q_2}{8\omega}\right)b^3 = \frac{(3\omega-\hat{\omega})\beta\gamma\sin\varphi}{2\omega^2(1\mp u_c)^2}. \tag{43}$$

The periodic solution can be expressed analytically as:

$$u = \pm u_c + \frac{2b^2 Q_1}{3\omega^2} + b\cos(\omega T + \varphi) - \frac{b^2 Q_1}{3\omega^2}\cos^2(\omega T + \varphi). \tag{44}$$

According to the characteristic solutions of Equation (42), it shows that the theoretical periodic solution expressed by Equation (44) will become unstable if:

$$\left(\omega - \hat{\omega} - \left(\frac{5Q_1^2}{12\omega^3} + \frac{9Q_2}{8\omega}\right)b^2\right)\left(\omega - \hat{\omega} + \left(\frac{5Q_1^2}{12\omega^3} + \frac{3Q_2}{8\omega}\right)b^2\right) \geq \frac{\mu^2}{4}. \tag{45}$$

Based on Equations (43)–(45), the evolution of the periodic solutions of system (3) with the amplitude of AC voltage when $\omega = 0.6$ is shown in Figure 6. Obviously, when $V_{AC}$ increases from 0, the two non-trivial equilibria lose their stability; instead, there are two periodic attractors coexisting. The amplitudes of the two periodic attractors increases with the amplitude of AC voltage. The coexistence of multiple periodic attractors can be attributed to the disturbance of the bistable non-trivial equilibria of the system (3) when $V_{AC} = 0$ V.

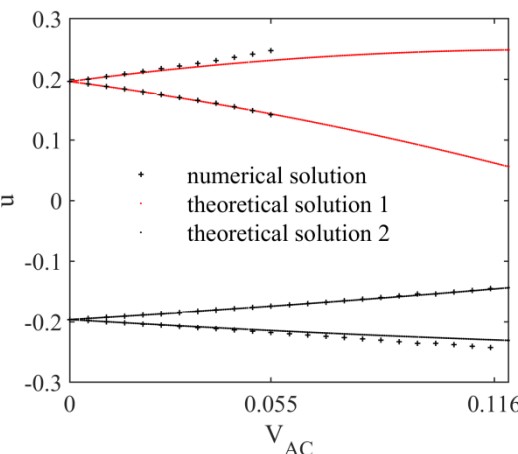

**Figure 6.** Variation of the periodic solutions with the amplitude of AC voltage when $\omega = 0.6$.

In Figure 6, when $V_{AC}$ varies from 0 to 0.055 V, the numerical simulation is in great agreement with the theoretical solution. However, when $V_{AC}$ exceeds 0.056 V, the theoretical prediction of the periodic attractor in the neighborhood of the right non-trivial equilibria is not that accurate, which may be due to the limitation of the Method of Multiple Scale. It will then be essential for us to apply numerical simulation to investigate the evolution of the attractors with the change in AC voltage. The basic settings for the simulation, such as the time step and initial plane, are the same as that in Section 3. The change of the attractors and the area and nature of their basins of attraction with $V_{AC}$ are shown in Figure 7, where $\omega = 0.6$. The evolution of global dynamics of system (3) with the increase in $V_{AC}$ can be separated into the following five stages.

Firstly, when $V_{AC} = 0$ V, there are two point attractors coexisting whose basins of attraction are fractal and trigger each other (see Figure 7a1,a2). According to Figure 7a2, in a small neighborhood of each point attractor, the attractor of system (3) is locally stable. Otherwise, a small disturbance of initial conditions will lead to a different point attractor, meaning that it is easy to induce a safe jump.

Secondly, when $V_{AC}$ increases from 0 to 0.01 V (see Figure 7b1–d2), the number of the periodic attractors increases with $V_{AC}$. At $V_{AC} = 0.005$ V, the two point attractors become two periodic attractors; apart from these two periodic attractors predicted theoretically, a new periodic attractor appears suddenly, marked by the yellow curve in Figure 7b1, and its basin of attraction is discrete (see the yellow regions in Figure 7b2). It shows that the new periodic attractor is a hidden attractor [26]. When $V_{AC}$ increases to 0.006 V, another hidden attractor appears, which is almost symmetric to the former one (see the blue curve of Figure 7c1 and the blue regions of Figure 7c2). When $V_{AC} = 0.01$ V, there are five periodic attractors coexisting, as shown in Figure 7d1. A new periodic attractor appears (see the green curve in Figure 7d1), whose amplitude is much bigger than the other ones.

Thirdly, as $V_{AC}$ increases from 0.01 V to 0.116 V, the number of attractors will decrease. Comparing Figure 7e1 with Figure 7d1, it is obvious that when $V_{AC}$ increases to 0.02 V, the yellow periodic attractor disappears whose basin of attraction is eroded by that of the green attractor; thus, the basin of attraction of the green attractor can be much bigger in Figure 7e2 than in Figure 7d2. When $V_{AC}$ continues to increase, the other three periodic attractors, i.e., the blue attractor, the red one, and the black one, disappear successively (see Figure 7f1,h1,j1) whose basins of attraction are aggressed by the basin of attraction of the green attractor, as shown in Figure 7e1–j2. Till $V_{AC}$ becomes 0.116 V, there will be a single periodic attractor left whose basin of attraction is not fractal but with a smooth boundary (see Figure 7j1,j2).

Besides, when $V_{AC}$ increases to 0.128 V, there will be a new complex attractor coexisting with the former green periodic attractor. It is a period-3 attractor (see the purple curve in Figure 7k1) whose basin of attraction is fractal and eroded to the basin of attraction of the periodic attractor (see Figure 7k2). It follows that a small change of initial conditions possibly shifts the dynamical behavior of the system (3) from a periodic motion to a period-3 motion, which is another type of safe jump.

Finally, as $V_{AC}$ continues to increase, another type of complex dynamical behavior is induced. According to the phase map, Poincare map, and frequency spectrum in Figure 8a–c, there is only a chaotic attractor when $V_{AC}$ = 0.28 V, and the boundary of its basin of attraction is not fractal (see Figure 8d).

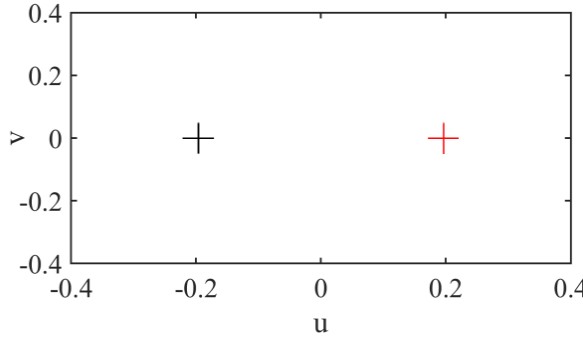

(**a1**) Attractors when $V_{AC}$ = 0 V

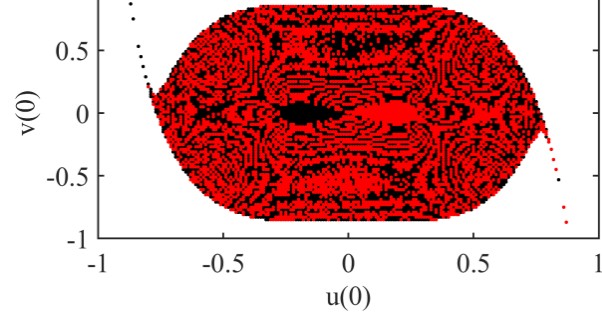

(**a2**) Basins of attraction when $V_{AC}$ = 0 V

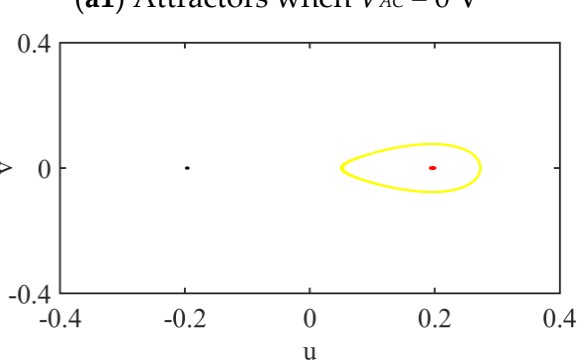

(**b1**) Attractors when $V_{AC}$ = 0.005 V

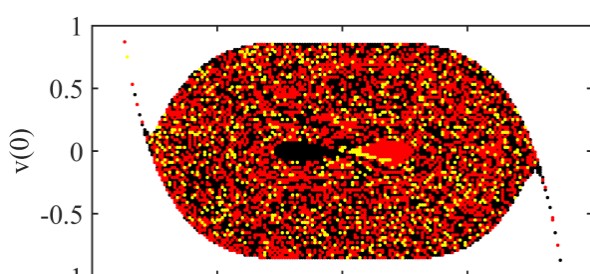

(**b2**) Basins of attraction when $V_{AC}$ = 0.005 V

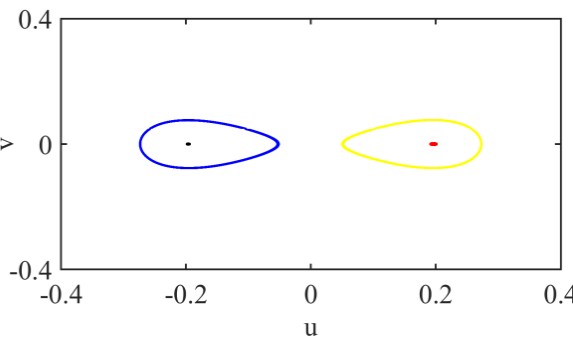

(**c1**) Attractors when $V_{AC}$ = 0.006 V

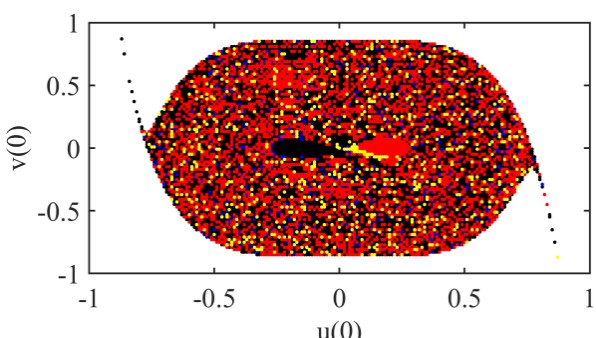

(**c2**) Basins of attraction when $V_{AC}$ = 0.006 V

**Figure 7.** *Cont.*

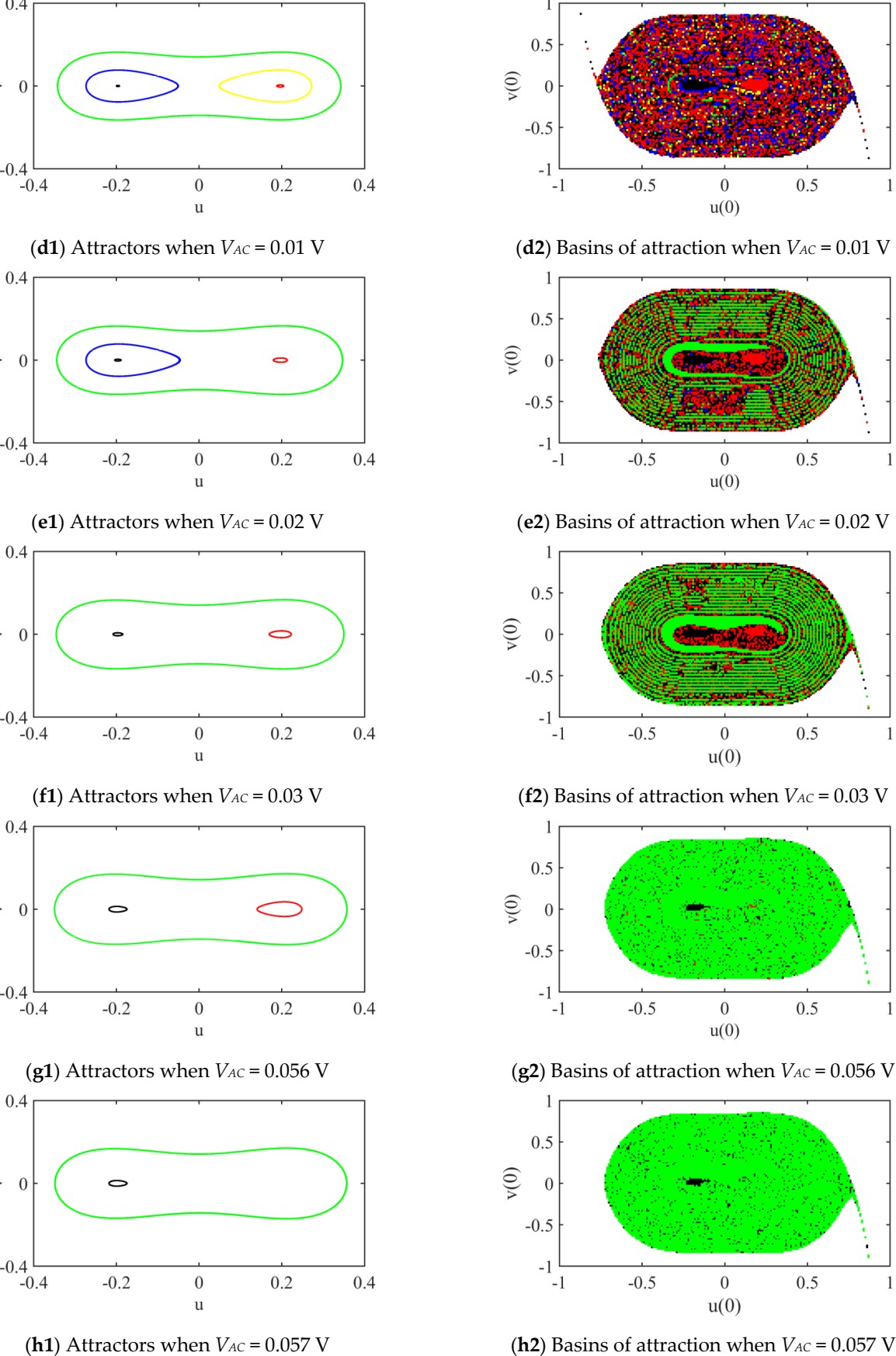

(**d1**) Attractors when $V_{AC}$ = 0.01 V

(**d2**) Basins of attraction when $V_{AC}$ = 0.01 V

(**e1**) Attractors when $V_{AC}$ = 0.02 V

(**e2**) Basins of attraction when $V_{AC}$ = 0.02 V

(**f1**) Attractors when $V_{AC}$ = 0.03 V

(**f2**) Basins of attraction when $V_{AC}$ = 0.03 V

(**g1**) Attractors when $V_{AC}$ = 0.056 V

(**g2**) Basins of attraction when $V_{AC}$ = 0.056 V

(**h1**) Attractors when $V_{AC}$ = 0.057 V

(**h2**) Basins of attraction when $V_{AC}$ = 0.057 V

**Figure 7.** *Cont.*

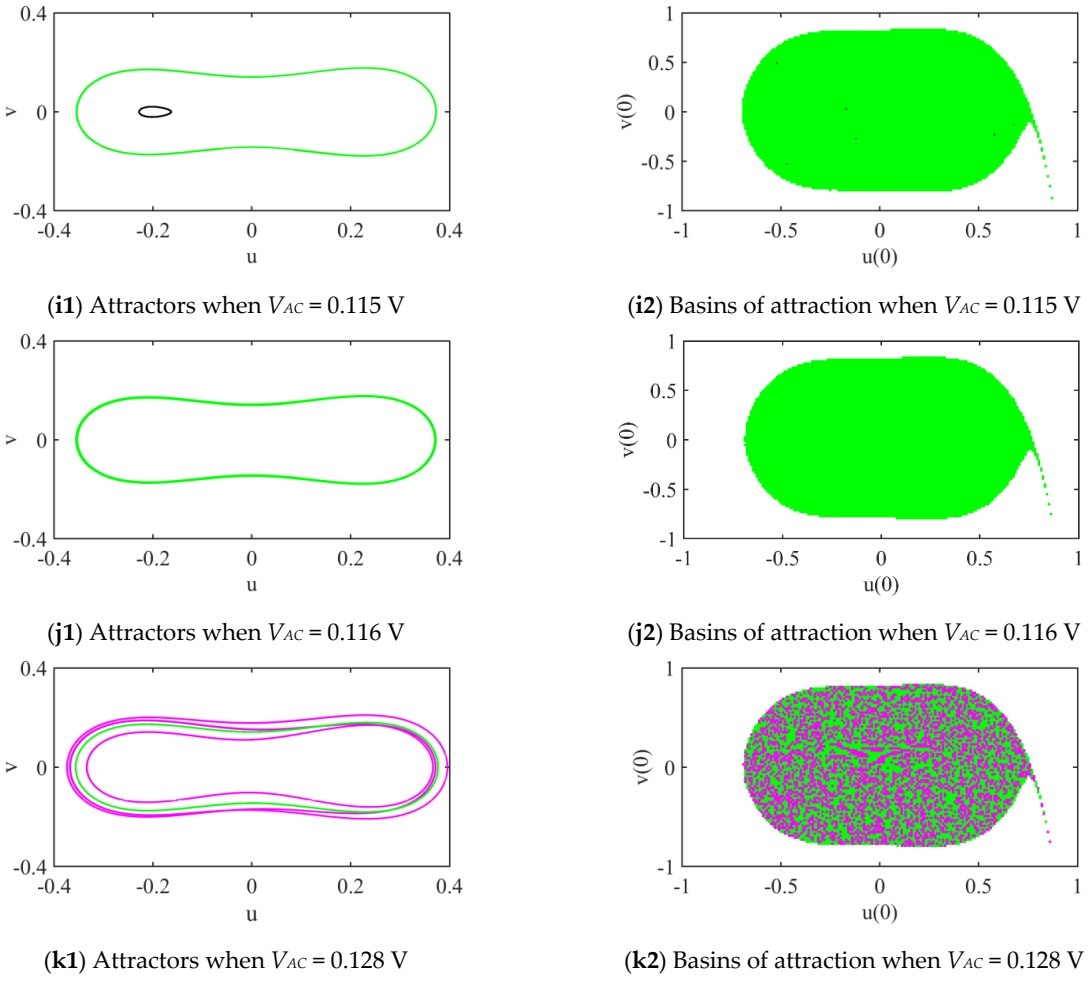

(**i1**) Attractors when $V_{AC}$ = 0.115 V                    (**i2**) Basins of attraction when $V_{AC}$ = 0.115 V

(**j1**) Attractors when $V_{AC}$ = 0.116 V                    (**j2**) Basins of attraction when $V_{AC}$ = 0.116 V

(**k1**) Attractors when $V_{AC}$ = 0.128 V                    (**k2**) Basins of attraction when $V_{AC}$ = 0.128 V

**Figure 7.** Evolution of multiple attractors and their basins of attraction under different values of $V_{AC}$.

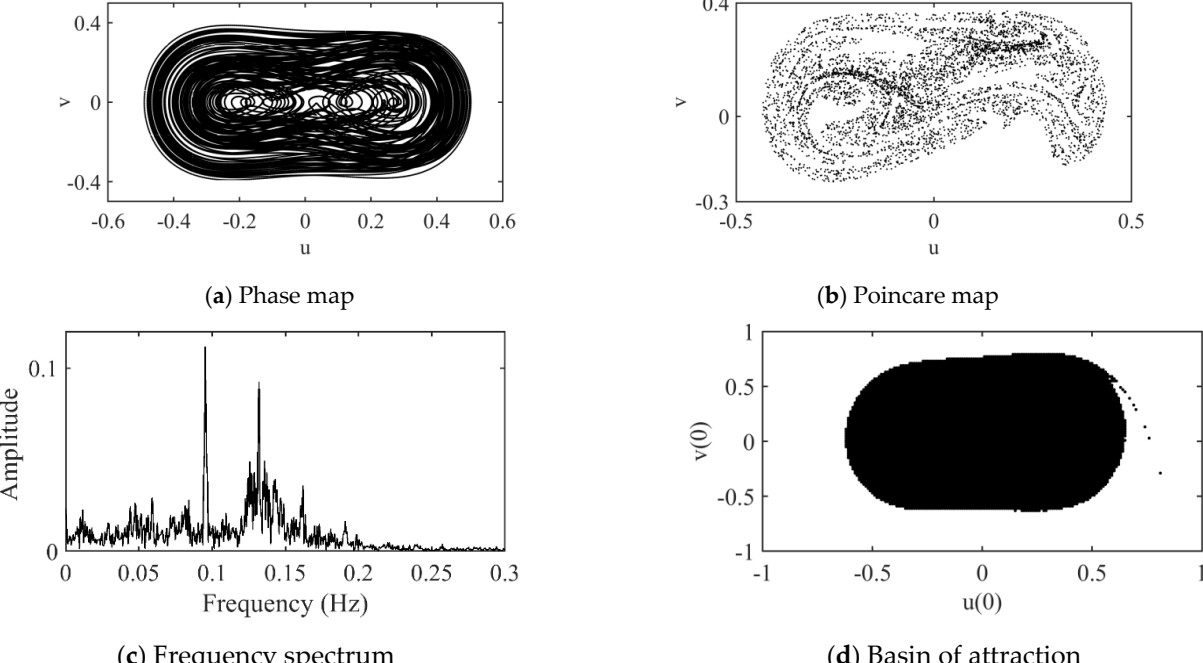

(**a**) Phase map                                                    (**b**) Poincare map

(**c**) Frequency spectrum                                      (**d**) Basin of attraction

**Figure 8.** Attractor and its basin of attraction when $V_{AC}$ = 0.28.

## 5. Conclusions

In this paper, a typical electrostatic bilateral micro-resonator is considered. The theory of local bifurcation and numerical approaches are applied to analyze the global dynamics of the vibrating system of the micro resonator. The main conclusions are presented as follows:

(1) DC bias voltage has some effect on the dynamics of the micro resonator. Without AC voltage, when the DC bias voltage is low, there will be only one stable point attractor in its vibrating system; when the DC bias voltage increases, there may be bistable point attractors.

(2) In the case of a low bias DC voltage, multiple periodic attractors and the corresponding safe jump occur due to Hopf bifurcation when varying the frequency or amplitude of AC voltage in certain ranges.

(3) Under a higher bias DC voltage that can induce bistable point attractors, when increasing the value of the amplitude of AC voltage, there will be multiple periodic attractors attributed to the loss of stability of the two non-trivial point attractors; apart from this, there will be some other complex dynamical behaviors of the micro-resonator vibrating system, such as safe jump, hidden attractors, period-n attractor, and chaos.

Our results provide some theoretical reference in avoiding complex dynamics of micro resonators, thus having some potential values in the design of micro sensors. The hidden attractors are depicted numerically, but their mechanism is still not that clear, which will be discussed in our future study.

**Author Contributions:** Conceptualization, H.S.; methodology, H.S.; software, Y.Z.; validation, H.S.; formal analysis, H.S.; investigation, Y.Z.; writing—original draft preparation, Y.Z. and H.S.; writing—review and editing, H.S.; visualization, Y.Z.; supervision, H.S.; project administration, H.S.; funding acquisition, H.S. All authors have read and agreed to the published version of the manuscript.

**Funding:** This research was funded by the National Natural Science Foundation of China, grant number 11472176.

**Institutional Review Board Statement:** Not applicable.

**Informed Consent Statement:** Not applicable.

**Data Availability Statement:** Not applicable.

**Acknowledgments:** Huilin Shang acknowledges the support of the National Natural Science Foundation of China under grant number 11472176. The authors are grateful for the valuable comments of the reviewers.

**Conflicts of Interest:** The authors declare no conflict of interest.

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
