# Peer review of "Multistability of the Vibrating System of a Micro Resonator"

_fractalfract, doi:10.3390/fractalfract6030141_

Round 1
Reviewer 1 Report
The work in this paper is done well. However, English language and style are not good enough to be published without revision. Refer to the file with highlighted comments to improve please. In addition, it may be much better for authors to re-number the subfigures (a), (b), (c), (d) et al in Fig. 7 as (a1), (a2), (b1), (b2) et al since every two subfigures correspond to same value of Vac in my opinion.

Author Response
Dear reviewer:
Thank you very much for your referee’s report. Based on your comments and suggestions, we have revised the manuscript. The corrected sentences or parts in the revised manuscript are highlighted in light blue for your convenience. Each of your questions was answered below.
- Abstract
Original
Firstly, the dynamical model is established and made dimensionless. Then, via the perturbating method and the numerical description of basins of attraction, the multiple periodic motions under primary resonance are discussed.
Revised
‘First, the dynamical model is established and made dimensionless. Second, via the perturbating method and the numerical description of basins of attraction, the multiple periodic motions under primary resonance are discussed.’
- Introduction
1) Original
Tt is of great significance to study the multistability and necessary conditions for inducing it either for avoiding this phenomenon or making use of it.
Revised
‘It is of great significance to study the multi-stability and necessary conditions for inducing it either for avoiding this phenomenon or making use of it.’
2) Original
To begin with, the dynamical model is constructed and made dimensionless.
Revised
‘The paper is organized as follows. In Section 2, the dynamical model is constructed and made dimensionless.
- Dynamic model
Original
where m represents the effective lumped mass of the moving electrode, k1 its linear mechanical stiffness k3 its cubic nonlinear stiffness, c the damping coefficient, C0 the initial capacitance of the parallel-plate structure.
Revised
‘where m represents the effective lumped mass of the moving electrode, k1 its linear mechanical stiffness, k2 its cubic nonlinear stiffness, c the damping coefficient, C0 the initial capacitance of the parallel-plate structure.
- Multiple Periodic Attractors in the Neighborhood of the Origin
1) Original
Figure 4 demonstrates that the parameters ω and VAC can induce the coexistence of bistability, meaning that under fixed values of parameters of the system (3), different initial conditions may lead to a different periodic motion.
Revised
‘Figure 4 demonstrates that the parameters ω and VAC can induce the coexistence of bistability, meaning that under fixed values of parameters of the system (3), different initial conditions may lead to different periodic attractors.’
2) Original
According to Figure 5, with the increase of ω, the number of attractors and the boundary of basins of attraction both changes.
Revised
According to Figure 5, with the increase of the parameter ω, the number of attractors and the boundary of basins of attraction will both change.
- Multistability in the Neighborhood of Non-trivial Equilibria
1) Original
Comparing coefficients Comparing the coefficients of and , one has
Revised
‘Comparing coefficients of and , one has’
2) Original
Substituting Equation (37) into Equation (35), and eliminating the secular terms of Equation (35), one will obtain that
Revised
‘Substituting Equation (37) into Equation (35), and eliminating the secular terms of Equation (35), one can obtain that’
3) Original
it may be much better for authors to re-number the subfigures (a), (b), (c), (d) et al in Fig. 7 as (a1), (a2), (b1), (b2) et al since every two subfigures correspond to same value of Vac.
A: 3) Thank you so much for your suggestion! Following this, our statement will be clearer. Thus, I have revised the corresponding parts in the manuscript which are highlighted in light blue color (see pages 7-8 and pages 10-13).
- Conclusions
Original
Firstly, the dynamical model is established and made dimensionless. Then, via the perturbating method and the numerical description of basins of attraction, the multiple periodic motions under primary resonance are discussed. It is found that the variation of AC voltage can induce safe jump of the micro resonator. Besides, with the increase of the amplitude of AC voltage, hidden attractors and chaos may appear
A: We deleted this paragraph, as we have briefly introduced the steps in Section 1 and the conclusions in Section 5.
Besides, we have checked the English expression in the entire manuscript. The corrected sentences or parts in the revised manuscript are highlighted in red color.
If you have any question, please contact us without hesitation. Thanks a lot!
Best regards,
Yijun Zhu, Huilin Shang

Reviewer 2 Report
In this paper, a typical electrostatic bilateral micro-resonator was considered by the authors.
I would like to suggest a paper to improve the state of the art ( adding comments).
- On suppression of chaotic motion of a nonlinear MEMS oscillator. Nonlinear Dyn99, 537–557 (2020). https://doi.org/10.1007/s11071-019-05421-8
Author Response
Dear reviewer:
Thanks for your referee’s report. Based on your comments, we have read the paper carefully and added it to our manuscript (highlighted in purple color for your convenience).
In the second paragraph of Section 1, we cited the paper you suggested as follows.
Angelo et al [24] investigated the effect of the linear and nonlinear stiffness terms and damping coefficients on dynamical behaviors of a microelectromechanical resonator and controlled the chaotic motion by forcing it into an orbit obtained analytically via the harmonic balance method.
And we revised the references correspondingly. The former references [24] and [25] now become [25] and [26], respectively.
[24] Angelo, M. T.; Jose, M. B.; Rodrigo, T. R.; et al. On suppression of chaotic motion of a nonlinear MEMS oscillator, Nonlinear Dyn. 2020, 99, 537-557.
Besides, we have checked the English expression in the entire manuscript. The corrected sentences or parts in the revised manuscript are highlighted in red color.
If you have any question, please contact us without hesitation. Thanks a lot!
Best regards,
Yijun Zhu, Huilin Shang

Reviewer 3 Report
Dear authors of the article, check the text of the article. There are a number of inaccuracies. For example, in formula (1) and further, the notation k2 is used for x^3. But the explanation of the formula describes k3.
Author Response
Dear reviewer:
Thank you very much for your referees’ reports. Based on your comments and suggestions, we have checked and revised the explanation of the parameters in formula (1) and corrected the formula (7). The corrected parts in the revised manuscript are highlighted in yellow color for your convenience.
For instance, the concerned part in Section 2 becomes:
Please see equation 1 in the article.
where m represents the effective lumped mass of the moving electrode, k1 its linear mechanical stiffness, k2 its cubic nonlinear stiffness…
Please see equation 7 in the article
Besides, we have checked the English expression in the entire manuscript. The corrected sentences or parts in the revised manuscript are highlighted in red color.
If you have any question, please contact us without hesitation. Thanks a lot!
Best regards,
Yijun Zhu, Huilin Shang
